# Cervical Joint Position Sense and Its Correlations with Postural Stability in Subjects with Fibromyalgia Syndrome

**DOI:** 10.3390/life12111817

**Published:** 2022-11-07

**Authors:** Ravi Shankar Reddy, Jaya Shanker Tedla, Snehil Dixit, Abdullah Raizah, Mohammed Lafi Al-Otaibi, Kumar Gular, Irshad Ahmad, Mohamed Sherif Sirajudeen

**Affiliations:** 1Department of Medical Rehabilitation Sciences, College of Applied Medical Sciences, King Khalid University, Abha 61421, Saudi Arabia; 2Department of Orthopedic Surgery, College of Medicine, King Khalid University, Abha 61421, Saudi Arabia; 3Department of Physical Therapy and Health Rehabilitation, College of Applied Medical Sciences, Majmaah University, Al Majmaah 11952, Saudi Arabia

**Keywords:** position sense, postural control, fibromyalgia, balance, proprioception

## Abstract

Cervical joint position sense (JPS) and postural stability are vital to maintaining balance and preventing falls in fibromyalgia syndrome (FMS). Impaired cervical JPS may influence postural stability, and understanding the relationship between them can aid in formulating treatment strategies in individuals with FMS. This study aims to (1) assess cervical JPS and postural stability between FMS and control group and (2) determine the correlation between cervical JPS and postural stability in the FMS group. This cross-sectional study recruited 92 FMS patients (mean age: 51.52 ± 7.7 yrs.) and 92 healthy controls (mean age: 49.36 ± 6.9 yrs.). A cervical range of motion (CROM) unit was utilized to assess cervical JPS. The postural stability was assessed using an IsoFree force platform, and anterior-posterior (A/P) and medial-lateral (M/L) directions of sway and ellipse area were measured. Cervical JPS and postural stability tests were assessed and compared between FMS and control groups. Cervical JPS was significantly impaired in FMS compared to the control group (*p* < 0.001). The JPS errors in FMS group were larger in flexion (FMS = 5.5°, control = 2.4°), extension (FMS = 6.4°, control = 3.0°), and rotation in left (FMS = 5.4°, control = 2.2°) and right directions (FMS = 5.1°, control = 2.8°). FMS individuals demonstrated statistically significant impaired postural stability compared to control in both the dominant and non-dominant legs tested (*p* < 0.001). The cervical JPS test showed moderate to strong positive correlations with postural stability variables. Statistically significant correlations were observed in all the JPS directions tested with all the postural stability variables (A/P and M/L sway and ellipse area). The correlation coefficients ranged between r = 0.37 (moderate) to 0.75 (strong). Cervical JPS and postural stability are impaired in FMS individuals. A moderate to strong relationship existed between JPS and postural stability. Individuals with FMS who had a greater magnitude of cervical JPS errors exhibited more severe postural control deficits. Therefore, cervical JPS and postural stability tests should be incorporated into routine clinical practice when assessing or formulating treatment strategies for patients with FMS.

## 1. Introduction

Fibromyalgia syndrome (FMS) is a chronic and long-term disorder characterized by widespread pain, fatigue, stiffness, cognitive decline, poor sleep, and psychosocial disabilities [1,2]. All of these complicated symptoms have a negative impact on quality of life, as well as on social and occupational functioning [3,4]. FMS is a prevalent problem that disproportionately affects women [5]. FMS affects between 0.2% and 6.6% of the population as a whole but between 2.6% and 6.8% of women [5]. Researchers suggest that fibromyalgia modifies the manner in which the brain and spinal cord receive painful and nonpainful signals, hence magnifying painful sensations [6].

Proprioception, referred to as kinaesthesia, is your body’s ability to sense movement, motion, and location [7,8]. Proprioception is required for precise and synchronized action planning, preserving static and dynamic balance, and optimal posture maintenance. In addition, it influences re-education and motor learning. [9,10]. Impaired position sense reduces the capacity to maintain joint stability and impairs postural control and bodily balance [11].

Cervical joint position sense (JPS) in the neck region can be evaluated using various techniques, such as the target head-positioning tests, ultrasound-based kinaesthetic evaluation systems, and three-dimensional FASTRAK measurement procedures [12]. Although it is not the gold standard, the target head-repositioning test employing a cervical range of motion (CROM) device is the most employed method in scientific research [9,13,14,15]. The multitude of receptors in the cervical joint, capsule, ligament, and peri-articular structures transmits afferent proprioceptive signals to the higher centers and contributes to efficient proprioception [16]. Neck pain intensity, muscle fatigue, and cervical injury have all been shown to affect joint position sense and measured as joint position error [17,18]. The neck’s target reposition test is a frequently adopted method to assess cervical JPS, and the test showed excellent reliability in individuals with and without neck pathology [19]. Alahmari et al. [19] demonstrated high intra-rater (intraclass correlation coefficient (ICC): 0.70 to 0.83) and inter-rater (ICC: 0.62 to 0.84) reliability of target head position assessments in subjects with neck pain and healthy individuals [10,19].

To conduct the necessary neuromuscular activity to maintain postural stability, one must integrate multidirectional sensory inputs (somatosensory, visual, and vestibular), motor, and cognitive inputs [20]. In a survey of 2596 FMS patients, postural control impairments were identified as among the most prevalent symptoms (45%) [21]. It is anticipated that clinical symptoms such as pain, sleep disturbances, physical fatigue, decreased muscle strength and endurance, impaired proprioception, anxiety, sensory deficits, and cognitive impairments may contribute to postural instability [21,22]. Chronic pain and fatigue are the primary symptoms of FMS [23]. Previous research indicates that several populations, including those with neck pain, proprioceptive, and postural disturbances, are associated with pain and muscle fatigue [24,25]. In addition, different authors showed that proprioceptive retraining enhanced postural control [26,27]. These findings suggest that cervical JPS and postural control are interdependent and can impact one another. However, chronic pain, reduced muscle strength, endurance, and perceived fatigue may significantly influence the severity of these effects in FMS patients. Limited studies assess different components of cervical JPS and postural control and their relationship in FMS. This study aims to (1) assess and compare cervical JPS and postural stability between FMS and the control (asymptomatic) group and (2) assess the relationship between cervical JPS and postural stability in the FMS group. 

## 2. Materials and Methods

### 2.1. Study Design, Participants, and Ethics

Ninety-two FMS patients (mean age: 51.52 ± 7.73 yrs.) and 92 healthy participants (mean age: 49.36 ± 6.93 yrs.) were recruited into this cross-sectional study using convenience sampling. This study was undertaken in the physical therapy clinics between August 2020 and January 2022. This study adhered to the Declaration of Helsinki, a declaration of moral guidelines for medical research involving human subjects, and this study was approved by the local university ethics committee (REC #2020-07-09). All subjects signed the consent form and voluntarily participated in this study. 

### 2.2. Inclusion and Exclusion Criteria

The FMS individuals were included in this project if they met the guidelines laid down by the American College of Rheumatology criteria (2010 guidelines) [28]. The FMS individuals were excluded if they had cervical spondylosis, cervical disc disease, a history of dizziness or vertigo, previous surgeries to the cervical spine, any balance and cognitive dysfunction, or a diagnosis of polyneuropathy or diabetes mellitus. All the participants went through a thorough physical examination by the physical therapist. Individuals with neck pain or muscle spasms and difficulty moving the neck were excluded from the study. In addition, complaints of pain, numbness, and the nature of radiating pain were assessed. Based on the patient’s medical history, radiological imaging techniques were used to examine individuals with suspected cervical injury or pathology. The individuals were excluded from the trial after the suspicion was confirmed. Patients taking tricyclic antidepressants, pregabalin, or serotonin noradrenaline reuptake inhibitors were excluded from the study since their use might influence the outcomes of this study.

The healthy individuals were included if they had no known health problems and were willing to participate voluntarily.

The demographics, widespread pain index, symptom severity score, duration of pain in years, Fibromyalgia impact questionnaire, pain catastrophizing, short-form 36 (SF-36), and exercise status were recorded for all individuals during the initial consultation.

### 2.3. Outcome Measures

#### 2.3.1. Cervical Joint Position Sense Testing

Cervical target head-position testing is a commonly used method to assess and estimate cervical JPS [13,19,29]. The participant relocates the head to the target from the neutral position, and reposition accuracy is noted as an absolute error [29]. An increase in absolute error is an indication of impaired cervical position sense. The examiner determines the target position, which is 50% of the total range of motion available in the neck. Cervical JPS is assessed with a cervical CROM device, often used to examine cervical proprioception [14,15,30]. The CROM unit is a mechanical device that incorporates conventional inclinometers and a magnetic reference to measure cervical ranges [31]. The test–retest reliability of cervical JPS testing with the CROM showed moderate-to-good reliability, with the intra-class correlation coefficient (ICC) ranging between 0.66 and 0.93 [15].

The subjects were instructed to sit comfortably with their backs straight on the chair and hands resting on the armrest. The examiner placed the CROM unit on the subject’s head and secured it. The magnetic yoke of the CROM unit is placed around the neck squarely, so that arrow of the magnetic yoke faces toward the north (Figure 1). 

The subject is instructed to close his eyes to ascertain his neutral head position, and the examiner calibrates the CROM unit to the zero position. From the neutral head position, the examiner directs the subject’s neck to the target position in flexion (50% of the available ROM), holds it there for 5 s, and asks the subject to memorize it. The examiner returns the subject’s head to the neutral or starting position. The subject is then instructed to move his head to the target position and indicates by saying “YES” when he has attained the desired position. The reposition accuracy is measured in degrees. The cervical JPS is measured in flexion, extension, and rotation in the left and right directions. Three trials were conducted in each movement direction, and the mean of these three trials is considered to quantify cervical JPS.

#### 2.3.2. Postural Stability Assessment

Each individual’s static postural stability was assessed as the individual stood on a single leg using a stabilometric force platform (IsoFree medical equipment, Tecnobody SRL, 2015). It has specialized software, an advanced force platform, a monitor, and a three-dimensional camera, and all work synchronously to provide real-time postural feedback (Figure 2). 

All postural control evaluations were conducted in a quiet, well-ventilated atmosphere. The participants were instructed to wear comfortable attire and stay relaxed. Prior to the actual test, there was a practice session to familiarize participants with the equipment. The examiner calibrated the IsoFree equipment before commencing the postural stability test. The individuals were instructed to stand barefoot in a monopodial stance in a standard position on the force platform. Next, the individuals were asked to flex the leg that was not being supported to avoid making contact with the force platform. After that, the participants were instructed to keep their balance by keeping their arms at their sides and focusing directly on the target mark displayed on the computer monitor. During each test, the individual was asked to stand on the testing leg for a total of 30 s; three trials are carried out on each leg, and the result that was deemed to be the most accurate was used for analysis. Anterior–posterior (A/P) and medial–lateral (M/L) directions of sway in millimeters and ellipse area were measured in mm^2^.

The cervical JPS and postural stability tests were administered by a physical therapist who was unaware of the individuals’ characteristics.

### 2.4. Statistical Analysis

The results of the Shapiro–Wilk test, which was used to confirm normality, were utilized to direct the analyses. Depending on the findings of the normality test, the independent t-test was used to compare the cervical JPS and postural control variables between FMS and asymptomatic individuals. The magnitude of the correlation between cervical JPS and postural control in FMS individuals was determined by the Pearson correlation coefficient (r). Between 0 and 0.3, correlation coefficients are regarded as weak, between 0.4 and 0.6 as moderate, and between 0.7 and 1.0 as strong. Multiple regression analysis was used to check the multicollinearity and shared inflation factors. Our criterion for acceptable multicollinearity among the factors was less than 10% of the common variance. The results were analyzed using SPSS version 24.0, and a probability level of 5% was deemed statistically significant. 

## 3. Results

The cross-sectional study included 92 FMS patients and 92 healthy participants. The physical characteristics of the patient and control groups did not differ significantly from one another (*p* > 0.05) and are summarized in Table 1.

In assessing cervical proprioception, the JPS errors were significantly larger (*p* < 0.001) in all directions (flexion, extension, left rotation, and right rotation) in the FMS group (Table 2). The results indicate that FMS patients have reduced cervical proprioception, as seen by more significant absolute errors while repositioning the neck to the target position. Compared to all movement directions, the magnitude of JPS errors was most significant in the extension direction in both FMS (6.4 ± 1.53°) and the control group (3.0 ± 1.25°). 

The results showed significantly decreased postural stability in the FMS group compared to the control group (*p* < 0.001) (Table 2). FMS participants demonstrated increased sway on both the dominant and non-dominant sides compared to the control group (Table 2).

In FMS patients, correlation analyses were undertaken between the cervical JPS and postural stability parameters, as summarized in Table 3. 

There was a statistically significant moderate to strong positive correlation between cervical JPS and postural stability parameters (*p* < 0.001). Furthermore, this significant correlation was seen between all the cervical JPS directions and all the postural stability variables tested. The findings imply that FMS patients with a larger magnitude of JPS errors have greater impaired postural control.

Including all the variables, the total amount of explained variance assessed by the variance inflation factor was only 3.05%, indicating the absence of multicollinearity. Furthermore, the observed tolerance was 54.6% above 0.10%, confirming the absence of multicollinearity.

## 4. Discussion

This cross-sectional aimed to compare cervical JPS and postural stability between FMS and control (asymptomatic) individuals and to assess the relationship between cervical JPS and postural stability in FMS individuals. The results indicate that cervical JPS and postural stability are impaired in the FMS group compared to the control group. In addition, a strong relationship existed between cervical JPS and postural stability variables in FMS individuals. 

Cervical sensorimotor control is vital for maintaining head position in space and controlling posture and balance. The findings of this study showed that cervical JPS is impaired in flexion, extension, and rotation in the left and right directions in FMS individuals. Our findings concur with previously published research indicating that FMS patients have impaired proprioception. Vaillant et al. [32] demonstrated that persons with FMS had absolute errors of >4.5°, whereas controls had 2.5°, indicating impaired position sense in FMS patients [32]. Celenay et al. [33] showed higher trunk reposition errors (>3.8°) in FMS patients compared to controls (*p* = 0.002) [33]. Contrary to our study results, Ulus et al. [34] showed no differences in knee proprioceptive acuity between FMS and controls [34]. FMS patients may have deteriorated proprioception for several reasons. Different authors have shown muscle pain, and perceived fatigue can impair proprioception [10,35,36]. Reddy et al. [36] demonstrated an impaired cervical-cephalic kinaesthetic sensibility following dorsal neck muscle fatigue [36]. Neck muscle fatigue alters the sensory receptors and the muscle spindles’ sensitivity, affecting the afferent proprioceptive input to the higher centers and ultimately impairing the cervical JPS [36,37]. Chronic pain can change the sensitivity of the muscle spindle receptors and alter afferent proprioceptive signals, impairing JPS [30]. Reduced proprioception has been associated with decreased activity in the deep muscle group, which contains the proprioceptive network [30]. Pain can affect the neurological system in several ways, including the sensitivity of the muscle spindles and the central nervous system’s response to cervical afferent inputs. Furthermore, the build-up of metabolites like potassium, lactic acid, and arachidonic acid might influence afferent proprioceptive input and impair cervical JPS [10,13,38,39]. The aforementioned chronic pain and fatigue mechanisms can explain proprioceptive deficits in FMS patients.

This study showed that postural sway was increased in FMS individuals, indicating reduced postural stability. In addition, moderate to strong positive correlations are observed between cervical JPS and postural stability values. Consequently, these findings show a relationship between impaired cervical JPS and poor balance. Previous studies have examined and documented balance impairments in FMS individuals and the effects of different exercise regimens that aim to improve postural stability [40,41]. Impaired postural control and risk of falling were identified in FMS individuals compared to controls, and the risk of falling was increased in individuals without taking medication [22]. Similar to our study results, Gucmen et al. [24] demonstrated a strong relationship between cervical repositioning tests and static and dynamic balance. Cervical JPS errors in flexion, extension, left and right rotations were correlated with the one-leg balance test and timed up-and-go test (*p* < 0.001) [24]. 

The A-P sway, M-L sway, and ellipse values were an average of 2.03 times higher in FMS group compared to asymptomatic. This result means that the center of pressure (COP) trajectory covers a larger area in this population in single-leg standing and it is consistent with the results of a previous study conducted in subjects with fibromyalgia who reported a similar increase in the area in a bipedal test with eyes open [42]. The tendency for the linear values of COP, sway area, and range of the COP to increase in FMS can be attributed to chronic pain experienced by this population [43]; it is possible that the motor pattern adapts to this condition and therefore modifies movement and stiffness to protect against further pain [44]. Previous studies in which pain was experimentally induced have reported a larger area [45]. This is attributed to the impact of pain on γ motor neuron activity [45], which could modulate the neuromuscular response. Additionally, impaired postural stability in FMS has been linked to reduced muscle strength and endurance, a decline in cognitive function, impairment in the stomato-sensory system, and pain-processing dysfunctions [46]. To effectively maintain body balance, the head should interpret the spatial orientation accurately. An extremely sensitive somatosensory system in the neck area collaborates with the visual and vestibular systems to provide precise proprioception. This system is crucial to the process of maintaining joint stability and bodily balance. [47]. Impaired cervical JPS may involve complex network interactions that alter the static or dynamic postural stability [48,49]. 

The results of this study have important clinical implications because the cervical JPS and postural control impairments are significant in FMS individuals compared to asymptomatic ones. Individuals with FMS should be assessed for cervical proprioceptive impairments in clinical practice. There is a strong relationship between cervical JPS and postural control, indicating that FMS individuals with impaired proprioception and postural control may have an increased frequency of falls [50,51]. Previous studies have shown improvements in postural control with proprioceptive exercises [52,53,54]. Therefore, rehabilitation therapists should consider improving cervical proprioception using different exercise regimes and assess their impact on postural control and the frequency of falls.

These results, in conjunction with those previously reported, reinforce the hypothesis of a disrupted somatosensory system in this population as the main cause of their altered postural stability. However, future studies should aim to provide further knowledge about proprioceptive information in these people using specific assessment tools. Furthermore, the results of this study should be taken cautiously because of fatigue, one of the most common symptoms of this population [55,56]. We only conducted three trials of each condition in the postural stability assessment. The number of trials should be expanded when the population assessed could perform the protocol without fatigue to improve the reliability of the tests. 

### Limitations

This study employed a common approach for assessing cervical JPS; however, it is not considered the gold standard. A “FASTTRACK” system with computer software can provide more specific information. In addition, the study evaluated absolute errors (global errors) only. Future research should examine constant and variable errors to obtain more information on the direction and quantity of errors in each direction of movement testing. Furthermore, because a cross-sectional study design was adopted, a causal association between the occurrence of a temporal sequence and disease progression could not be demonstrated. However, the first stage of planning a longitudinal study could be the inferences from a cross-sectional study. We evaluated three trials of cervical JPS testing in each movement direction, while previous authors used six to ten trials in each direction [52]. We considered three attempts since more repetitions may cause hip-muscle fatigue, alter afferent proprioceptive input, and impair JPS [57,58]. In addition, exercise status among FMS is highly variable, and this factor may influence FMS pain severity, cervical JPS, and postural control. Future research evaluating cervical JPS and postural stability in this cohort should consider these factors.

## 5. Conclusions

The study’s findings indicated that cervical JPS and postural stability are impaired in the FMS group compared to controls. There was a significant relationship between cervical JPS and postural stability, suggesting that patients with a larger magnitude of proprioceptive errors had increased postural sway values and impaired postural stability. Cervical JPS and postural stability evaluations may be implemented in routine clinical practice when evaluating or formulating treatment strategies in FMS patients. As cervical proprioception and postural control are essential for preventing falls and a substantial correlation has been seen between them, future rehabilitation research should assess the effectiveness of these aspects in preventing falls in FMS patients.

## Figures and Tables

**Figure 1 life-12-01817-f001:**
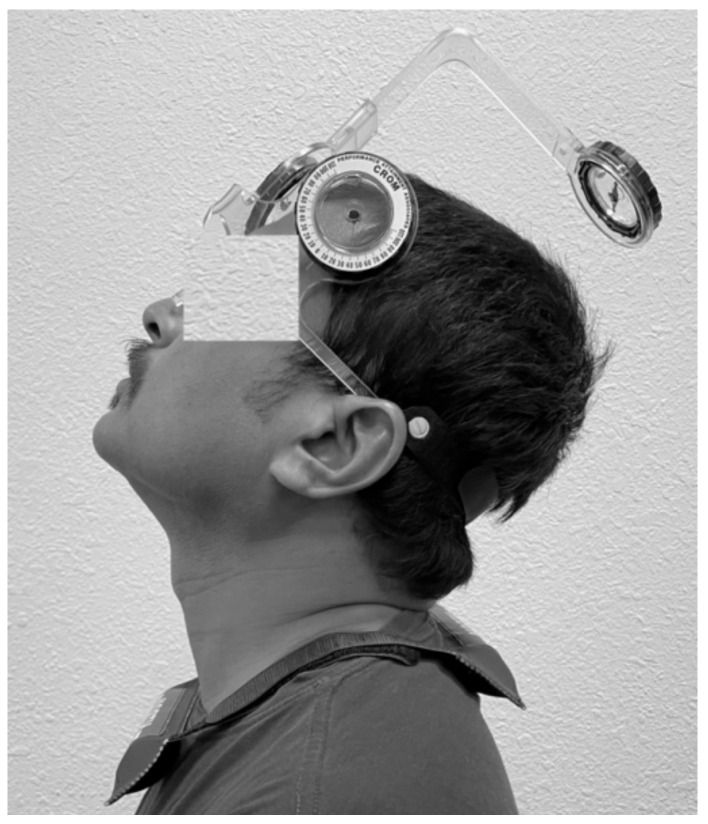
Cervical joint position sense testing using a CROM device.

**Figure 2 life-12-01817-f002:**
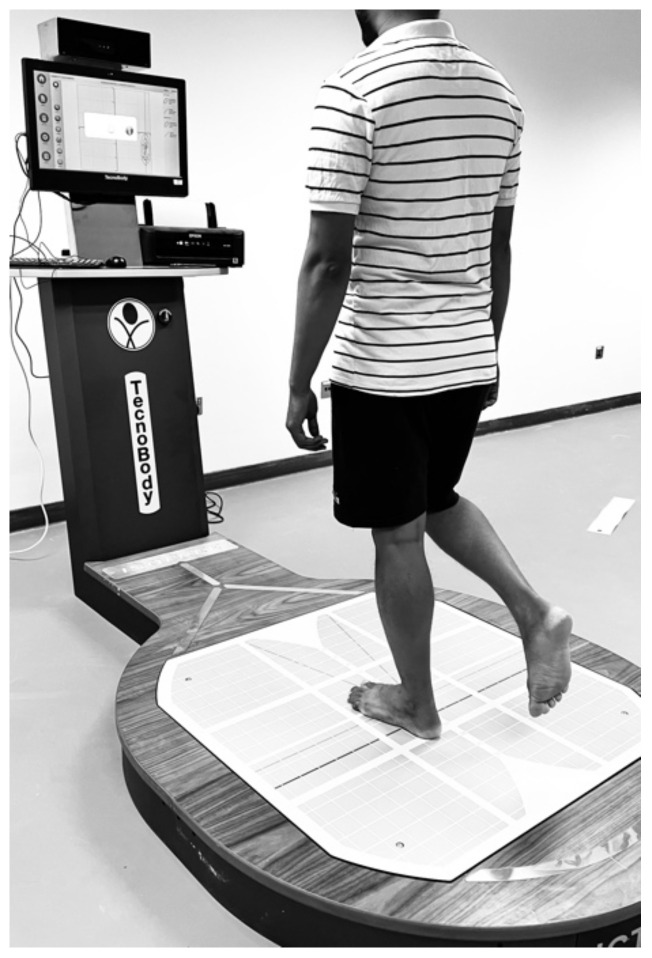
Postural stability assessment using a stabilometric force platform.

**Table 1 life-12-01817-t001:** Physical and demographic characteristics of participants.

Variables	FMS Individuals (n = 92)	Asymptomatic (n = 92)	*p*-Value
Age (years)	51.52 ± 7.7	49.36 ± 6.9	0.267
Gender (M: F)	39:53	27:65	0.010
Height (meters)	1.68 ± 0.10	1.63 ± 0.05	0.521
Weight (kg)	72.20 ± 6.44	69.73 ± 5.23	0.052
BMI (kg/m^2^)	25.72 ± 4.02	25.38 ± 2.81	0.621
Pain intensity: VAS (0–10 cm)	5.9 ± 1.3	-	-
Widespread pain index	13.90 ± 2.50	-	-
Symptom severity score	9.86 ± 1.70	-	-
Duration of pain in years	6.82 ± 2.43	-	-
Fibromyalgia impact questionnaire	61.10 ± 10.0	-	-
Pain catastrophizing	30.53 ± 4.96	-	-
SF-36	37.01 ± 9.02	-	-
Exercise status (n)		-	-
Never	36		
1–2 days/week	28		
3–4 days/week	16		
>4 days/week	12		

FMS = fibromyalgia syndrome, BMI = body mass index, VAS = visual analogue scale.

**Table 2 life-12-01817-t002:** Comparisons of cervical JPS and postural stability variables between FMS and asymptomatic individuals.

Variables	FMS Group (n = 92) Mean ± SD	Asymptomatic Group (n = 92) Mean ± SD	*p*-Value
Cervical JPS in flexion (°)	5.5 ± 1.43	2.4 ± 1.30	<0.001
Cervical JPS in extension (°)	6.4 ± 1.53	3.0 ± 1.25	<0.001
Cervical JPS in rotation left (°)	5.4 ± 1.36	2.9 ± 0.79	<0.001
Cervical JPS in rotation right (°)	5.1 ± 1.29	2.8 ± 0.63	<0.001
Anterior–posterior sway (mm)—non-dominant	9.92 ± 3.65	3.32 ± 1.20	<0.001
Medial–lateral sway (mm)—non-dominant	7.34 ± 2.45	3.88 ± 1.55	<0.001
Ellipse Area (mm^2^)—non-dominant	986.2 ± 152.63	457.88 ± 151.83	<0.001
Anterior-posterior sway (mm)—dominant	8.65 ± 3.34	3.35 ± 1.27	<0.001
Medial–lateral sway (mm)—dominant	6.15 ± 1.83	3.49 ± 1.58	<0.001
Ellipse area (mm^2^)—dominant	966.91 ± 136.09	432.30 ± 154.40	<0.001

JPS= joint position sense, FMS= fibromyalgia syndrome.

**Table 3 life-12-01817-t003:** Coefficient of correlation between cervical JPS and postural stability variables in FMS individuals.

Variables		Cervical JPS in Flexion (°)	Cervical JPS in Extension (°)	Cervical JPS in Rotation Left (°)	Cervical JPS in Rotation Right (°)	Anterior-Posterior Sway (mm)—Non-Dominant	Medial–Lateral Sway (mm)—Non-Dominant	Ellipse Area (mm^2^)—Non-Dominant	Anterior-Posterior Sway (mm)—Dominant	Medial–Lateral Sway (mm)—Dominant]	Ellipse area (mm^2^)—Dominant
Cervical JPS in flexion (°)	r	1									
Cervical JPS in extension (°)	r	0.575 **	1								
Cervical JPS in rotation left (°)	r	0.591 **	0.557 **	1							
Cervical JPS in rotation right (°)	r	0.586 **	0.546 **	0.801 **	1						
Anterior-posterior sway (mm)–non-dominant	r	0.613	0.573	0.562	0.552	1					
Medial–lateral sway (mm)—non-dominant	r	0.574 **	0.377 **	0.526 **	0.545 **	0.531 **	1				
Ellipse area (mm^2^)—non-dominant	r	0.650 **	0.642 **	0.667 **	0.651 **	0.645 **	0.617 **	1			
Anterior–posterior sway (mm)—dominant	r	0.543 **	0.483 **	0.524 **	0.548 **	0.514 **	0.583 **	0.705 **	1		
Medial–lateral sway (mm)—dominant	r	0.511 **	0.460 **	0.438 **	0.441 **	0.469 **	0.531 **	0.559 **	0.591 **	1	
Ellipse area (mm^2^)—dominant	r	0.649 **	0.662 **	0.646 **	0.585 **	0.713 **	0.614 **	0.761 **	0.683 **	0.606 **	1

JPS = joint position sense, FMS = fibromyalgia syndrome, ** = correlation is significant at the 0.01 level.

## Data Availability

The data used for this investigation have been uploaded as “Raw data” under the Appendix A.

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
