# Peer review of "Cervical Joint Position Sense and Its Correlations with Postural Stability in Subjects with Fibromyalgia Syndrome"

_life, 2022, doi:10.3390/life12111817_

Round 1

Reviewer 1 Report

The manuscript is presented in a well-structured manner, it is clear and well-written. The topic is original and very interesting both from a medical and rehabilitation point of view. The results are clearly presented and well-related to the conclusions. The figures are appropiate and the tables are easy to interpret and they properly show the data.

The cited references are relevant to the research, but can be improved. Some of these are not so recent and there are too many self-references

Author Response

Response to Reviewer comments

Thank you for your effort and time in reviewing our manuscript. The reviewing process has significantly improved the quality of this manuscript. Therefore, I am submitting this "Response to reviewers" document summarizing the changes we made in response to the critiques. In addition, I have highlighted the changes manuscript.

Reviewer 1

Sl.no

Queries

Response to queries

1.      

 The manuscript is presented in a well-structured manner, it is clear and well-written. The topic is original and very interesting both from a medical and rehabilitation point of view. The results are clearly presented and well-related to the conclusions. The figures are appropriate and the tables are easy to interpret and they properly show the data.

Thank you for the comments.

2.      

The cited references are relevant to the research, but can be improved. Some of these are not so recent.

The cited references are reviewed and improved as per your suggestion.

The references are updated with recently published articles.

Reviewer 2 Report

THE WORK OF REMARKABLE SCIENTIFIC INTEREST, SHOWS AND CERTIFIES CERTAIN IDEAS, UNDERLINING THE INITIAL LOGICAL DEDUCTIONS WITH HIGHLY VALIDATED AND PROVEN DATA. THE METHODS OF MEASUREMENT ARE OF HIGH QUALITY AND SPECIFIC, THE STATISTICAL STUDY WELL STRUCTURED, THE STUDY METHODOLOGY PRECISE AND ALSO ATTENTIVE TO PRACTICAL ERRORS THAT COULD INDICATE FALSE. THE RESEARCH MATERIALS INDICATE A VALID NUMBER OF PATIENTS, GOOD AGE RANGE AND GOOD TEST MOTORS OF STIMULUS USED.

Author Response

Response to Reviewer comments

Thank you for your effort and time in reviewing our manuscript. The reviewing process has significantly improved the quality of this manuscript. Therefore, I am submitting this "Response to reviewers" document summarizing the changes we made in response to the critiques. In addition, I have highlighted the changes manuscript.

Reviewer 2

Sl.no

Queries

Response to queries

1.      

THE WORK OF REMARKABLE SCIENTIFIC INTEREST, SHOWS AND CERTIFIES CERTAIN IDEAS, UNDERLINING THE INITIAL LOGICAL DEDUCTIONS WITH HIGHLY VALIDATED AND PROVEN DATA. THE METHODS OF MEASUREMENT ARE OF HIGH QUALITY AND SPECIFIC, THE STATISTICAL STUDY WELL STRUCTURED, THE STUDY METHODOLOGY PRECISE AND ALSO ATTENTIVE TO PRACTICAL ERRORS THAT COULD INDICATE FALSE. THE RESEARCH MATERIALS INDICATE A VALID NUMBER OF PATIENTS, GOOD AGE RANGE AND GOOD TEST MOTORS OF STIMULUS USED.

Thank you for your comments.

Reviewer 3 Report

Thank you for your invitation to revise this manuscript. I regret to give a negative feedback this time because of the tools used in this study. 

Regarding the JPSE, the authors reported a detailed description of the methodology used. However, a mean average of 6 trials (instead of 3) is needed to obtain reliable measurements (doi:10.1016/j.jmpt.2022.08.005). In addition, the utility of this test is highly controversial (doi:10.1093/ptj/pzz167) since it shows the lowest reliability and unsatisfactory construct validity (doi:10.1186/1471-2474-15-408).

Regarding the force platform, its utility is also highly controversial (doi.org/10.3390/s22062365). For instance, a previous study controlling more external and internal factors influencing the balance than this study described an error in the measurements up to the 60% for the ML displacement (possibly due to the high intra-subject variability). 

Additionally, how can a AP or a ML sway be reported in mm2 if this is a linear displacement?

Regarding the statistical analyses I would recommend the authors revising some concerns. Regarding the correlation analyses, I would recommend to use a correlation matrix for assessing collinearity and shared variance instead of pair correlations as is. Also, the exercise status is highly variable among the cases group. This should be considered as each status may influence in FMS severity, proprioception and body balance. 

Author Response

Response to Reviewer comments

Thank you for your effort and time in reviewing our manuscript. The reviewing process has significantly improved the quality of this manuscript. Therefore, I am submitting this "Response to reviewers" document summarizing the changes we made in response to the critiques. In addition, I have highlighted the changes manuscript.

Reviewer 3

Sl.no

Queries

Response to queries

1.      

Thank you for your invitation to revise this manuscript. I regret to give a negative feedback this time because of the tools used in this study. 

Regarding the JPSE, the authors reported a detailed description of the methodology used. However, a mean average of 6 trials (instead of 3) is needed to obtain reliable measurements (doi:10.1016/j.jmpt.2022.08.005).

·       We considered three attempts since more repetitions may cause muscles fatigue, alter afferent proprioceptive input, and further impair proprioception.

·        We have experience evaluating cervical proprioception, and during our clinical investigations, we have observed that increasing the number of repetitions in symptomatic individuals leads to a greater magnitude of proprioceptive errors, and this can significantly change the results.

·        There are number of studies published assessing cervical proprioception using three trials

o   Alshahrani, A., Aly, S.M., Abdrabo, M.S. and Asiri, F.Y., 2018. Impact of smartphone usage on cervical proprioception and balance in healthy adults. Biomed res29(12), pp.2547-2552.

o   Alahmari K, Reddy RS, Silvian P, Ahmad I, Nagaraj V, Mahtab M. Intra-and inter-rater reliability of neutral head position and target head position tests in patients with and without neck pain. Brazilian journal of physical therapy. 2017 Jul 1;21(4):259-67.

o   Swait, G., Rushton, A.B., Miall, R.C. and Newell, D., 2007. Evaluation of cervical proprioceptive function: optimizing protocols and comparison between tests in normal subjects. Spine32(24), pp.E692-E701.

o   Reddy, R.S., Tedla, J.S., Dixit, S. and Abohashrh, M., 2019. Cervical proprioception and its relationship with neck pain intensity in subjects with cervical spondylosis. BMC musculoskeletal disorders20(1), pp.1-7.

o   Reddy, Ravi Shankar, Ney Meziat-Filho, Arthur Sá Ferreira, Jaya Shanker Tedla, Praveen Kumar Kandakurti, and Venkata Nagaraj Kakaraparthi. "Comparison of neck extensor muscle endurance and cervical proprioception between asymptomatic individuals and patients with chronic neck pain." Journal of Bodywork and Movement Therapies 26 (2021): 180-186.

o   Reddy RS, Meziat-Filho N, Ferreira AS, Tedla JS, Kandakurti PK, Kakaraparthi VN. Comparison of neck extensor muscle endurance and cervical proprioception between asymptomatic individuals and patients with chronic neck pain. Journal of Bodywork and Movement Therapies. 2021 Apr 1;26:180-6.

o   Alahmari, K.A., Reddy, R.S., Tedla, J.S., Samuel, P.S., Kakaraparthi, V.N., Rengaramanujam, K. and Ahmed, I., 2020. The effect of Kinesio taping on cervical proprioception in athletes with mechanical neck pain—a placebo-controlled trial. BMC musculoskeletal disorders21(1), pp.1-9.

o   Ulutatar, F., Unal-Ulutatar, C. and Duruoz, M.T., 2019. Cervical proprioceptive impairment in patients with rheumatoid arthritis. Rheumatology International39(12), pp.2043-2051.

o   Farley, T., Barry, E., Bester, K., Barbero, A., Thoroughgood, J., De Medici, A., Sylvester, R. and Wilson, M.G., 2022. Poor cervical proprioception as a risk factor for concussion in professional male rugby union players. Physical Therapy in Sport55, pp.211-217.

·        Taking into consideration we have mentioned as one of the limitations.

2.      

In addition, the utility of this test is highly controversial (doi:10.1093/ptj/pzz167) since it shows the lowest reliability and unsatisfactory construct validity (doi:10.1186/1471-2474-15-408).

·        Previous studies which evaluated intra-inter-rater reliability of cervical joint position sense showed good reliability.

o   Alahmari K, Reddy RS, Silvian P, Ahmad I, Nagaraj V, Mahtab M. Intra-and inter-rater reliability of neutral head position and target head position tests in patients with and without neck pain. Brazilian journal of physical therapy. 2017 Jul 1;21(4):259-67.

·        The studies methods you have mentioned (doi:10.1093/ptj/pzz167,  doi:10.1186/1471-2474-15-408) used to assess the cervical joint position sense are different from our study. Both cannot be compared.

3.      

Regarding the force platform, its utility is also highly controversial (doi.org/10.3390/s22062365). For instance, a previous study controlling more external and internal factors influencing the balance than this study described an error in the measurements up to the 60% for the ML displacement (possibly due to the high intra-subject variability). 

o   The force platform used in the Calvo-Moreno et al study is a basic version. We have used advanced computerized motion Analysis and Force Platform to record and check correct movements, balance and posture. Both cannot be compared.

o   Calvo-Moreno et al study was aimed to analyze how different environmental acoustic conditions affect the test–retest reliability and to analyze the most appropriate number of trials to calculate a valid mean average score. This study cannot be compared to our study.

4.      

Additionally, how can a AP or a ML sway be reported in mm2 if this is a linear displacement?

o   Sorry for the typo error. The errors are rectified.

5.      

Regarding the statistical analyses I would recommend the authors revising some concerns. Regarding the correlation analyses, I would recommend to use a correlation matrix for assessing collinearity and shared variance instead of pair correlations as is. Also, the exercise status is highly variable among the cases group. This should be considered as each status may influence in FMS severity, proprioception and body balance. 

-correlation matrix is now used for assessing collinearity and shared variance.

o   correlation matrix is now provided as table 3.

We have mentioned exercise status as one of the limitations, as exercise status is highly variable and may have an influence on FMS pain severity, cervical JPS, and postural control.
